# Machine-learning and mechanistic modeling of metastatic breast cancer after neoadjuvant treatment

**Sebastien Benzekry**[1]*, **Michalis Mastri**[2], **Chiara Nicolò**[3], **John M. L. Ebos**[2,4]

1 Computational Pharmacology and Clinical Oncology (COMPO), Inria Sophia Antipolis–Méditerranée, Cancer Research Center of Marseille, Inserm UMR1068, CNRS UMR7258, Aix Marseille University UM105, Marseille, France, 2 Department of Cancer Genetics and Genomics, Roswell Park Comprehensive Cancer Center, Buffalo, New York, United States of America, 3 InSilicoTrials Technologies S.P.A, Riva Grumula, Trieste, Italy, 4 Department of Medicine, Roswell Park Comprehensive Cancer Center, Buffalo, New York, United States of America

* sebastien.benzekry@inria.fr

**Data Availability Statement:** The data used in this study is publicly available at https://zenodo.org/records/10607753 [16]. The repository contains pre-surgical primary tumor volume measurements and pre- and post-surgical records of metastatic

## Abstract

Clinical trials involving systemic neoadjuvant treatments in breast cancer aim to shrink tumors before surgery while simultaneously allowing for controlled evaluation of biomarkers, toxicity, and suppression of distant (occult) metastatic disease. Yet neoadjuvant clinical trials are rarely preceded by preclinical testing involving neoadjuvant treatment, surgery, and post-surgery monitoring of the disease. Here we used a mouse model of spontaneous metastasis occurring after surgical removal of orthotopically implanted primary tumors to develop a predictive mathematical model of neoadjuvant treatment response to sunitinib, a receptor tyrosine kinase inhibitor (RTKI). Treatment outcomes were used to validate a novel mathematical kinetics-pharmacodynamics model predictive of perioperative disease progression. Longitudinal measurements of presurgical primary tumor size and postsurgical metastatic burden were compiled using 128 mice receiving variable neoadjuvant treatment doses and schedules (released publicly at https://zenodo.org/records/10607753). A non-linear mixed-effects modeling approach quantified inter-animal variabilities in metastatic dynamics and survival, and machine-learning algorithms were applied to investigate the significance of several biomarkers at resection as predictors of individual kinetics. Biomarkers included circulating tumor- and immune-based cells (circulating tumor cells and myeloid-derived suppressor cells) as well as immunohistochemical tumor proteins (CD31 and Ki67). Our computational simulations show that neoadjuvant RTKI treatment inhibits primary tumor growth but has little efficacy in preventing (micro)-metastatic disease progression after surgery and treatment cessation. Machine learning algorithms that included support vector machines, random forests, and artificial neural networks, confirmed a lack of definitive biomarkers, which shows the value of preclinical modeling studies to identify potential failures that should be avoided clinically.

burden in 251 mice implanted with human breast cancer cells either untreated or pre-surgically treated with two distinct receptor tyrosine kinase inhibitors (Sunitinib and Axitinib) and multiple dose and scheduling regimen. It also contains tumor and circulating biomarkers collected at surgery. The code is available at https://gitlab.inria.fr/benzekry/metamats.burden.treatment. A software heritage identifier permalink for a code snapshot at the time of publication is available at https://archive.softwareheritage.org/swh:1:snp:d39660e34a46938dba993b474433a38b7c51a70f;origin=https://gitlab.inria.fr/benzekry/metamats.burden.treatment [17].

**Funding:** This work used shared resources supported by the Roswell Park Comprehensive Cancer Center (RPCCC) Support Grant from the National Cancer Institute (NCI) (P30CA016056), to JMLE. This work was also supported by grants to JMLE from the American Cancer Society (ACS) via a Research Scholar Grant (RSG-18-064-01-TBG) and from the Roswell Park Alliance Foundation (RPAF). This work was also supported by Inria funding within the associated teams grant METAMATS, to SB. MM received salary from RPCCC. CN received salary from Inria. Opinions, interpretations, conclusions and recommendations are those of the author and not necessarily endorsed by the NCI, ACS, RPAF, or Inria. The funders had no role in study design, data collection and analysis, decision to publish, or preparation of the manuscript.

**Competing interests:** The authors have declared that no competing interests exist.

## Author summary

Using simulations from a mechanistic mathematical model compared with preclinical data from surgical metastasis models, we found that anti-tumor effects of neoadjuvant receptor tyrosine kinase inhibitor treatment can differ between the primary tumor and metastases in the perioperative setting. Model simulations with variable drug doses and scheduling of neoadjuvant treatment revealed a contrasting impact on initial primary tumor debulking and metastatic outcomes long after treatment has stopped. Using machine-learning algorithms, we identified the limited power of several circulating cellular and molecular biomarkers in predicting metastatic outcomes, uncovering a potential fast-track strategy for assessing future clinical biomarkers by pairing patient studies with identical studies in mice.

## Introduction

Neoadjuvant trials in breast cancer (BC) patients involve the administration of systemic treatment for a limited period to treat (and reduce) localized primary tumors before surgery [1]. They provide several advantages to assist in novel drug development and translational research. They can be faster to conduct (than, e.g., adjuvant trials), require fewer patients, offer the potential for controlled assessment of biological tissue for novel biomarker development, and critically, can potentially limit distant (often occult) metastatic lesions to delay disease recurrence long after treatment has ended [1,2]. Yet there are surprisingly few studies that precede neoadjuvant trial design to offer predictive guides to validate drug efficacy, biomarkers, or possible outcomes. In this regard, *in silico* (mathematical) modeling and preclinical *in vivo* testing can be useful. However, mathematical modeling most often occurs as post-hoc analysis in BC trials, and studies in mice rarely include clinically relevant systems that capture the complexity of surgical impact on primary/metastatic growth to offer rationalized inclusion of biomarkers in trial design [3].

To address this gap, here we describe a mathematical modeling framework of neoadjuvant therapy, using a combination of preclinical *in vivo* and *in silico* data. This extends from our prior work that validated a semi-mechanistic model comparing localized 'primary' tumor growth with the growth of spontaneous metastatic disease after surgery in mouse models of BC [4]. We used 'ortho-surgical' models (i.e., orthotopic implantation followed by surgical tumor resection) to show that inter-individual variability in the kinetics of metastatic growth could be captured by the distribution of a critical parameter of metastatic aggressiveness. However, critically, this previous mathematical model did not account for systemic treatment.

In the current study, we examined neoadjuvant treatment with sunitinib, a molecular targeted tyrosine inhibitor (TKI) that can block angiogenesis-associated vascular endothelial growth factor receptors (VEGFRs) along with several other regulators of metastasis [5]. Using a VEGFR TKI had several advantages. First, they have a short half-life, which allowed us to confine treatment effects to the presurgical period and incorporate multiple variations of treatment dosing, and duration or time of resection after initial tumor implantation in mice [6]. Second, VEGFR TKIs have shown mixed effects in the perioperative setting in BC [7]. While the addition of neoadjuvant sunitinib to chemotherapy improved pathologic complete response rates, long-term results have been more contrasted, with no disease-free survival benefit and either none [8] or some [9] overall benefit. As a monotherapy, we and others have demonstrated that robust inhibition of primary tumor growth does not always translate into inhibition of metastasis post-surgically nor improvement in survival [10,11]. However, the

mechanistic determinants of these counter-intuitive findings remain elusive. In the current experimental model, we measured multiple cellular and molecular biomarkers at surgery. Adding here neoadjuvant treatment to our mathematical modeling framework allowed us to 1) formulate and test mechanistic hypotheses about differential effects on primary versus secondary disease, 2) evaluate the impact of biomarkers on metastatic development and 3) investigate the impact of modulating dosing regimen.

For 2), we use machine learning to investigate the predictive power of biomarkers on the mechanistic parameters of our neoadjuvant mathematical metastatic model. Machine learning (ML) coupled with mechanistic modeling–an approach that we term 'mechanistic learning' [12,13]–can screen biomarkers with translational potential and establish predictive models [14]. In contrast to classical statistical analysis, ML consists in designing models with predictive performances as metrics of success, instead of inference properties. It also makes use of nonlinear models such as regression trees or artificial neural networks [15].

## Materials and methods

### Ethics statement

Animal studies were performed in strict accordance with the recommendations in the Guide for Care and Use of Laboratory Animals of the National Institute of Health and according to guidelines of the Institutional Animal Care and Use Committee at Roswell Park Comprehensive Cancer Center (protocol: 1227M, PI: John M.L. Ebos).

### Experimental system

**Cell lines.**   The human LM2-4$^{LUC+}$ cells are a luciferase-expressing metastatic variant of the MDA-MB-231 breast cancer cell line derived after multiple rounds of *in vivo* lung metastasis selection in mice, as previously described [5]. LM2-4$^{LUC+}$ were maintained in DMEM (Corning cellgro; 10-013-CV) supplemented with 5% v/v FBS (Corning cellgro; 35-010-CV), in a humidified incubator at 37oC and 5% CO2. The cell line was authenticated by STR profiling (DDC Medical, USA).

**Drug and doses used.**   Sunitinib malate (SU11248; Sutent, Pfizer) is a molecular receptor tyrosine kinase inhibitor (RTKI) that can block angiogenesis-associated vascular endothelial growth factor receptors (VEGFRs) along with several other regulators of metastasis [18]. The molecule was suspended in a vehicle formulation that contained carboxymethylcellulose sodium (USP, 0.5% w/v), NaCl (USP, 1.8% w/v), Tween-80 (NF, 0.4% w/v), benzyl alcohol (NF, 0.9% w/v), and reverse osmosis deionized water (added to final volume), which was then adjusted to pH 6. The drug was administered at 60 or 120 mg/kg/day orally by gavage as previously described [10,19]. The treatment window used in all neoadjuvant studies consisted of a previously optimized 14-day period before surgery [10]. Within these 14 days, daily sunitinib (Su) treatment was given either at 60 mg/kg/day (for 3, 7, or 14 days followed by vehicle for 11, 7, or 0 days, respectively), or at 120 mg/kg/day for 3 days followed by 60 mg/kg/day for 0, 4, 8, or 11 days, and vehicle for 11, 7, 3, or 0 days, respectively. An example of an abbreviation in the text includes 'Su60(14D)', which means 'sunitinib at 60mg/kg/day for 14 days. Schematics for all studies are shown in S1 Table. Mice treated daily with vehicle for 14 days were used as controls. Detailed analysis and comparisons of these treatment regimens are described in a companion study evaluating treatment breaks on metastatic disease.

**Ortho-surgical model of metastasis.**   *Implantations.* Experimental methodology was extended from previous work using a xenograft animal model of breast cancer spontaneous metastasis that includes orthotopic implantation followed by surgical resection of a primary tumor (termed 'ortho-surgical') [4]. Briefly, LM2-4$^{LUC+}$ (1 x 10$^6$ cells in 100μl DMEM) were

orthotopically implanted into the right inguinal mammary fat pad (right flank) of 6- to 8-week-old female SCID mice, as described previously [4,10,19]. Primary tumor (PT) burden was monitored with Vernier calipers using the formula width$^2$(length x 0.5) and bioluminescence imaging (BLI) [4,10,19]. Neoadjuvant treatments started 14 days before primary tumors were surgically removed at a timepoint (34–38 days post-implantation) previously optimized for maximal distant metastatic seeding but minimal localized invasion [4,10]. The surgeries were planned at specific time points post-implantation to avoid invasion of the primary tumor into the skin or peritoneal wall, ensuring that metastatic progression had proceeded and minimizing the possibility of surgical cure [4,10]. Postsurgical metastatic burden (MB) was assessed by BLI and overall survival was monitored based on signs of end-stage disease as previously described [4,10].

*Exclusion criteria.* Two scenarios represented instances where animals were excluded from treatment studies. First, if complete removal of the primary tumor was not surgically feasible because of local invasion or evidence of advanced metastatic spread [4,19]. Second, if no primary or metastatic tumor was ever detected by BLI or visual assessment it was assumed there was lack of tumor-take upon implantation [4,10].

*Randomization.* Before treatment initiation animals were randomized by primary tumor size assessed by Vernier calipers to avoid any false results due to unequal tumor burden between groups [20].

## Biomarkers

**Flow cytometry.**    Peripheral blood was collected in tubes containing lithium heparin (BD Biosciences; 365965) by orbital bleeding one day before surgical tumor resection. Non-specific binding was blocked with normal mouse IgG (Invitrogen; 10400C) incubated with whole blood, followed by incubation with an antibody mix. After staining, cells were fixed in a lyse/fix solution (BD Biosciences; 558049), while red blood cells were lysed. Samples were analyzed with a LSR II flow cytometer (Becton Dickinson), while data were acquired with FACSDiva software (Becton Dickinson) and analyzed with FCS Express 6 (DeNovo software).

**Circulating tumor cells (CTC).**    The antibody mix for CTC detection of human CTCs in animal models contained a rat anti-mouse CD45 (30-F11) antibody conjugated to PE (Biolegend; 103106) and mouse anti-human HLA conjugated to AlexaFluor 647 (Biolegend; 311416). CD45 staining with a rat anti-mouse CD45 conjugated to FITC (Invitrogen; MCD4501) was used to eliminate any mouse blood cells, whereas human HLA was used to identify CTC (human LM2-4$^{LUC+}$). For positive control, LM2-4$^{LUC+}$ cells were trypsinized, washed with PBS, and stained for both CD45 and HLA. LM2-4$^{LUC+}$ cells were used to define the CTC gate.

**Circulating myeloid-derived suppressor cells (MDSC).**    The antibody mix for detection of MDSCs contained a rat anti-mouse CD45 (30-F11) antibody conjugated to PE (Biolegend; 103106), a rat anti-mouse Ly-6G/Ly-6C (Gr1) (RB6-8C5) antibody conjugated to PE-Cy7 (BD Pharmingen; 552985), and a rat anti-mouse CD11b (M1/70) antibody conjugated to eFluor450 (eBioscience; 48–0112). Mouse CD45 staining was used to select only leukocytes, and CD11b and Gr1 were used to define the granulocytic and monocytic MDSC.

**Immunofluorescence.**    Resected tumors were frozen on dry ice in a cryo-embedding compound (Ted Pella, Inc; 27300), sectioned, and fixed in a 3:1 mixture of acetone:ethanol. Non-specific binding was blocked with 2% BSA in PBS, followed by staining with an antibody mix containing rabbit anti-mouse Ki67 antibody (Cell Signaling Technologies; 12202) and rat anti-mouse CD31 antibody (Dianova; DIA-310). Detection of primary antibodies was achieved using FITC conjugated goat anti-rabbit IgG (BD Pharmingen; 554020) and Cy3 conjugated

goat anti-rat IgG (Invitrogen; A10522). Samples were counterstained with DAPI (Vector; H-1500) and mounted with a hard-set mounting medium for fluorescence. Random images from each section were obtained with a Zeiss AxioImager A2 epifluorescence microscope at 200x magnification, and analyzed with ImageJ. CD31+ cells (% area) and Ki67+ cells (% cells) were quantified automatically using macro functions, whereas Ki67+/CD31+ cells (proliferating endothelial cells) were quantified manually.

## Mechanistic modeling of untreated pre- and post-surgical metastatic development

For untreated animals, we previously validated a mechanistic model of pre-surgical primary tumor growth and post-surgical metastasis development in an ortho-surgical LM2-4[LUC+] animal model [4]. Briefly, metastatic development is decomposed into two main processes: growth and dissemination.

Growth of the PT and metastases follow the Gomp-Exp model [21]:

$$g_p(v) = g(v) = \min\left(\lambda v, \left(\alpha - \beta ln\left(\frac{v}{V_0}\right)\right)v\right),$$

where $g_p$ and $g$ denote the growth rates of the primary and secondary tumors, respectively. Parameter $\lambda$ limits the Gompertz growth rate to avoid unrealistically fast kinetics for small sizes and is given by the *in vitro* proliferation rate, assessed previously [4]. Parameters $\alpha$ and $\beta$ are the Gompertz parameters, and $V_0$ is the size of one cell (in units of mm$^3$ for the PT, and photons/seconds for the metastases). The PT volume, $V_p(t)$ thus solves

$$\begin{cases} \dfrac{dV_p}{dt} = g_p\left(V_p\right) \\ V_p(t=0) = V_i, \end{cases}$$

with $V_i$ the volume corresponding to the number of cells injected (= 1 mm$^3$ based on the conversion rule 1 mm$^3 \simeq 10^6$ cells [22]).

Dissemination occurs at the following volume-dependent rate [4]:

$$d(V_p) = \mu V_p,$$

where parameter $\mu$ can be interpreted as the daily probability that a cell from the PT successfully establishes a metastasis [4].

The metastatic process was described through a function $\rho(t, v)$ representing the distribution of metastatic tumors with size $v$ at time $t$. It solves the following initial boundary value problem [23]:

$$\begin{cases} \partial_t\rho(t,v) + \partial_v(g(v)\rho(t,v)) = 0, & t \in (0 + \infty), v \in (V_0, +\infty) \\ g(t, V_0)\rho(t, V_0) = d(V_p(t)), & t \in (0, +\infty) \\ \rho(0, v) = 0, & v \in (V_0, +\infty) \end{cases}$$

The first equation derives from a balance equation on the number of metastases; the second equation is a boundary condition for the rate of newly created metastases; the third equation is the initial condition (no metastases at the initial time).

The metastatic burden (MB) at time $t$ was then given by

$$M(t) = \int_{V_0}^{+\infty} v\rho(t, v)dv = \int_0^t d(V_p(t - s))V(s)ds,$$

which can be solved efficiently through the use of a fast Fourier transform algorithm [24]. In the previous equation, $V(s)$ represents the volume reached by a metastatic tumor after time $s$ from its birth, when growing with growth rate $g$.

## Heuristic modeling of neoadjuvant therapy (NAT)

In the first phase of investigations (Fig 1), we leveraged the previously identified parameters of our model for untreated metastatic development [4]. NAT was grossly modeled by simply setting either $g_p$ only (Scenario A) or both $g_p$ and $g$ (Scenario B) to zero, during NAT. This allowed us to perform pure predictions that did not require any parameter fitting to the data.

## Kinetics-pharmacodynamics modeling of NAT

**Structural model.** We next incorporated the effects of systemic therapy in a way that accounts for systemic concentration kinetics resulting from the drug scheduling regimen (dose and timing of administration). This new model assumes that the drug reduces the primary tumor growth rate by a term proportional to its concentration, $C(t)$ (Norton-Simon hypothesis [25]):

$$g_p^T(t, v) = g_p(v)(1 - k\,C(t))$$

where $k$ is a parameter of drug efficacy. As no pharmacokinetic data was available, we used a kinetics-pharmacodynamics (K-PD) approach. Namely, we considered that the drug concentration decays exponentially after each dose,

$$C(t) = \frac{1}{V_d} \sum_{i=1}^n D_i e^{-k_e(t-\tau_i)} \mathbb{I}_{t > \tau_i},$$

where $D_i$ indicates the dose administered at time $\tau_i$. The pharmacokinetics parameters volume of distribution $V_d$ and the elimination rate constant $k_e$ were fixed to the values reported in [26].

**Parameter identification for the K-PD model.** Following previously established methodology [4], the parameters were identified from the experimental data using a nonlinear-mixed effects modeling approach [27]. Briefly, this consists of modeling inter-animal variability by assuming a parametric distribution for the model parameters. All individual PT and MB longitudinal data eated or not, were pooled together in a population model whose parameters were estimated by likelihood maximization [27]. In mathematical terms, if $y_j^i$ denotes the observation (PT or MB in animal $i$ at time $t_j^i$ and $f(t_j^i; \theta^i)$ denotes the model prediction in an animal with parameter set $\theta^i = (\alpha^i, \beta^i, k^i, \mu^i)$, we log-transformed the data and assumed a proportional error model. The statistical observation model thus writes:

$$\ln y_j^i = \ln(f(t_j^i; \theta^i))(1 + \bar\sigma \varepsilon_j^i),$$

$$\ln(\theta^i) = \ln(\theta_{pop}) + \eta^i, \eta^i \sim \mathcal{N}(0, \Omega),$$

where $\varepsilon_j^i \sim \mathcal{N}(0, 1)$ is a Gaussian noise for measurement error. The parameters $\theta_{pop}$ and $\Omega$ characterize the entire population. The observed data were log-transformed and a proportional

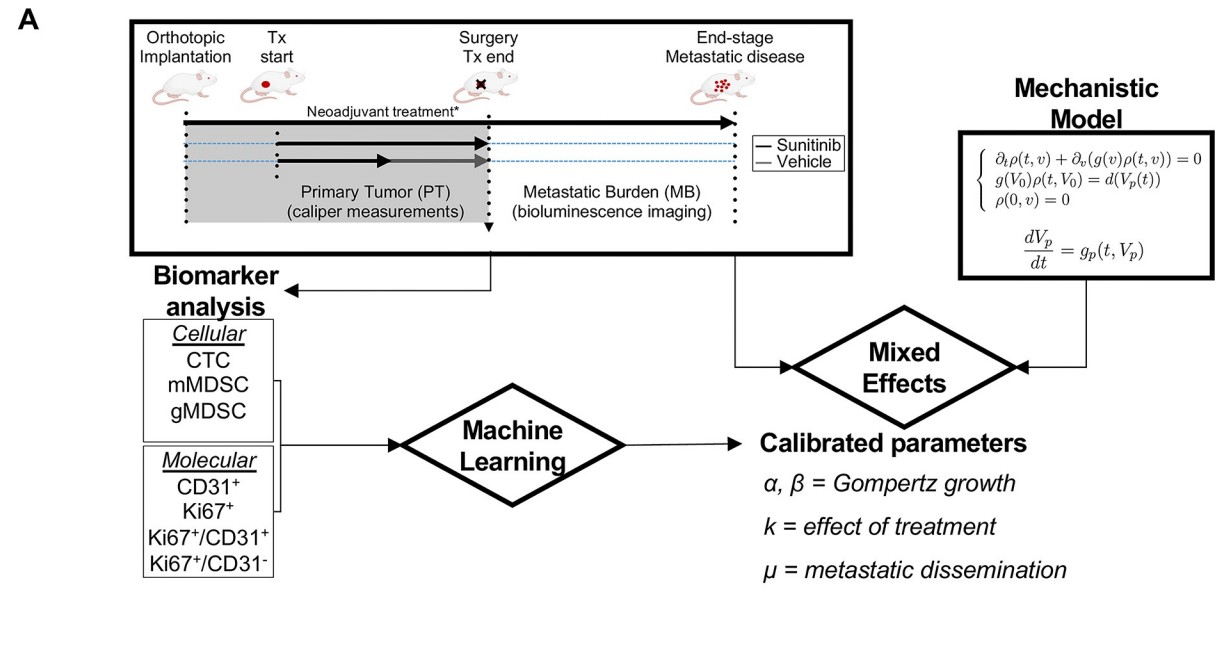

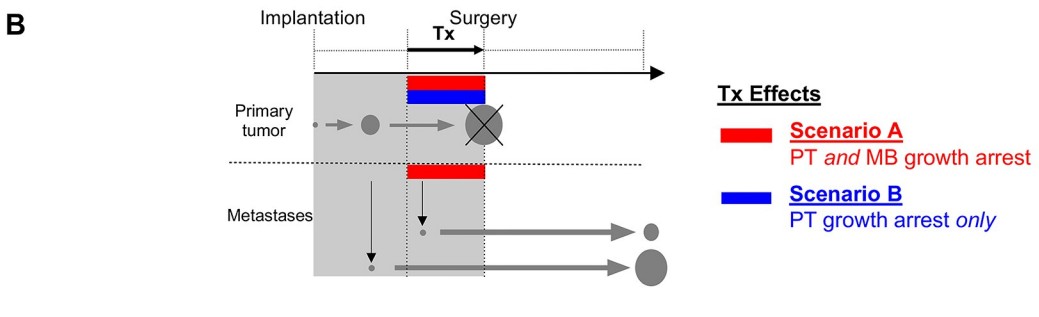

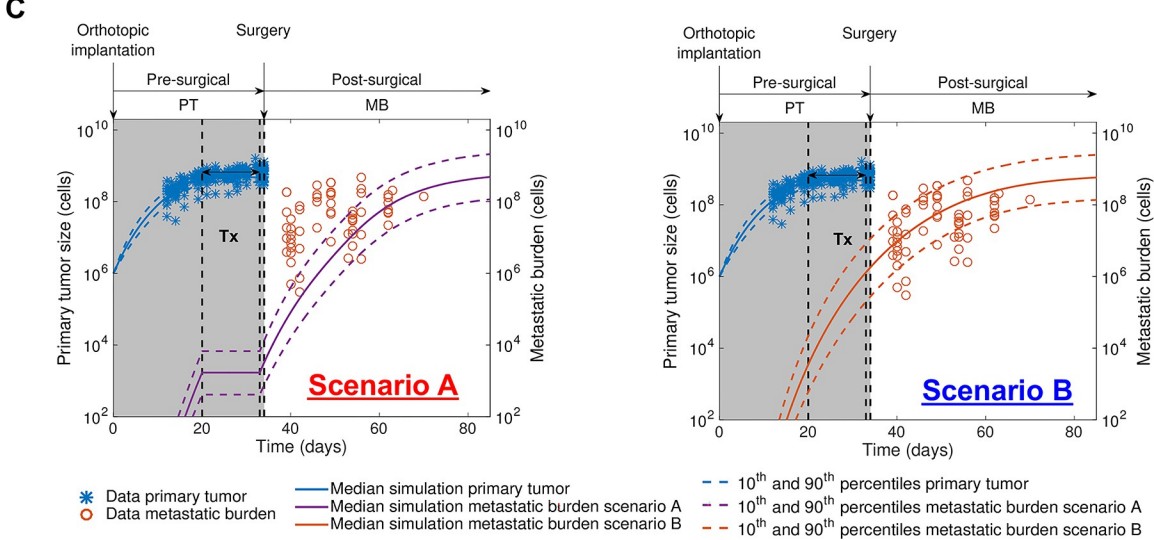

**Fig 1. Mathematical modeling reveals differential effects of neoadjuvant sunitinib treatment on primary tumor and metastatic growth.**
**(A)** Schematic of the study. Data from an ortho-surgical, human xenograft animal model of neoadjuvant sunitinib breast cancer treatment were fitted using a mixed-effects statistical framework. This provided calibrated parameters for each animal. Machine learning algorithms were used to assess the predictive power of molecular and cellular biomarkers to predict the metastatic dissemination parameter μ and quantify metastatic aggressiveness. **(B)** Schematic of tested hypotheses of the effect of neoadjuvant sunitinib Tx on primary tumor and

metastatic growth and dissemination through mechanistic mathematical modeling. Scenario A = growth arrest on both primary and secondary tumors. Scenario B = growth arrest on primary tumor only. **(C)** Predicted simulations of Scenarios A and B using parameters calibrated from a previous study involving untreated (vehicle) animals only [4]. Data plotted here (LM2-4[LUC+] bioluminescent human breast cancer cells orthotopically injected in mice) was not used to estimate the model parameters. *Tx, treatment; PT, primary tumor; MB, metastatic burden. *See* methods *for additional details on animal experiments, treatment dose and duration, and mechanistic model. The mouse images were drawn using Biorender.*

error model was used, that is

$$\ln y_j^i = \ln(f(t_j^i; \theta^i))(1 + \bar{\sigma}\varepsilon_j^i).$$

For the vector of individual parameters, a log-normal distribution with full covariance matrix was assumed. Maximum likelihood estimates of the population parameters were obtained using the Stochastic Approximation of Expectation-Maximization (SAEM) algorithm implemented in the *nlmefitsa* Matlab function [28]. PT and MB data were fitted simultaneously for vehicle and sunitinib-treated animals. Visual predictive checks (VPC), individual fits and standard diagnostic graphical tools based on individual parameters were used to evaluate the adequacy of the different model components.

**Definition of metastatic relapse.**   To define a time of metastatic relapse, we mimicked the human situation in which distant metastatic relapse occurs in 30% of breast cancer patients with localized disease at diagnosis [29]. Therefore, we set a MB relapse threshold to be the 30[th] percentile of the control population MB at 85 days (considered to be an approximation of long-term), i.e. $6.322 \times 10^8$ cells. This threshold then allowed us to compute the percent of subjects having metastatic relapse in the virtual populations, under varying NAT scheduling regimens.

## Machine learning algorithms

Effects of covariates on the model parameters were assessed using linear regression and several machine learning regression techniques (partial least squares, artificial neural networks, support vector machines and random forest models) using the R caret package [30,31]. Except for the random forest models, data were centered and scaled before modeling. Hyperparameters were tuned to minimize the root mean squared error (RMSE) using five replicates of a 10-fold cross-validation. If $\theta^i$ are the true values and $\hat{\theta}^i$ the predicted ones, the RMSE is defined by:

$$RMSE = \sqrt{\frac{\sum_i |\hat{\theta}^i - \theta^i|^2}{N}}.$$

## Results

### *In vivo/in silico* modeling of neoadjuvant therapy (NAT) suggests limited effect on metastasis growth

The previously reported "paradoxical" observation of a differential effect of NAT on the presurgical primary tumor (PT) and post-surgical metastatic burden (MB) [10,11] could result from two phenomena that are mixed in MB quantification: 1) metastatic growth suppression and 2) reduction of metastatic spread as a consequence of primary tumor size reduction. To disentangle the two, we used quantitative mathematical modeling to compare two biological hypotheses against the data. (Schematic shown in Fig 1A; see Methods for the mathematical model definition). Since our previously developed model did not account for treatment [4], we

first used a heuristic approach to model NAT. In 'scenario A', NAT was assumed to have growth-arresting effects on both PT and MB, while in 'scenario B' NAT was assumed to have an effect only on PT (schematically shown in Fig 1B). This way, we could use previously calibrated parameters [4], and only set either both growth rates $g_p$ and $g$ (Scenario 'A') or $g_p$ only (Scenario 'B') to zero during NAT. This allowed to make predictions without relying on data fitting. Scenario 'A' clearly failed to describe the data (Figs 1C and S1), whereas Scenario 'B' interestingly demonstrated good accuracy (Figs 1C and S1). These results demonstrate a differential effect of NAT on the growth of primary and secondary tumors and suggest that a mathematical model of NAT in our breast cancer ortho-surgical animal model should not include an anti-growth effect on metastasis.

## Calibration and validation of a kinetics-pharmacodynamics (K-PD) model for NAT

Subsequently, to further link dose and scheduling to response, we developed a K-PD metastatic model of NAT using experimental data with multiple treatment periods (3, 7, 11, 14 days), doses (60 mg and 120 mg), and time of surgery after tumor implantation (day 34 or 38) (see S1 Table and methods for details). Following our findings above, we only adapted the PT growth rate $g_p$ from [4], using the Norton-Simon hypothesis for the PT anti-growth effect of NAT [25]. Estimates of the model parameters are reported in Table 1 and demonstrate excellent practical identifiability (all relative standard errors $\leq$ 17%), likely owing to the large number of subjects in the population fit. In addition, the estimate of the proportional error parameter $\bar{\sigma}$ (3.91%) indicated accurate goodness-of-fit.

Confirming our previous results [4], the metastatic potential parameter $\mu$ was found to vary significantly amongst individuals (largest coefficient of variation). Visual predictive checks for both the vehicle group and treated groups demonstrated accurate goodness-of-fit both at the population (Figs 2A and S2) and individual (Figs 2B and S3) levels. In addition, model predictions in independent data sets unused for parameter calibration, with distinct times of surgery (day 38 versus day 34) and drug regimens, were in good agreement with the data (S4 Fig). Further model diagnostic plots demonstrated no clear misspecification of the structural and residual error models (S5 Fig). Distributions of the empirical Bayes estimates were in agreement with the theoretical distributions defined in the statistical model (S6 Fig). Moreover, the $\eta$-shrinkage was less than 20% for each parameter, indicating that the individual parameter estimates and the diagnostic tools based on them can be considered reliable [32]. Finally,

**Table 1. Parameter estimates of the metastatic and survival models obtained by likelihood maximization via the SAEM algorithm.**

| Parameter (Unit) | Meaning | Median value | CV (%) | r.s.e. (%) |
|---|---|---|---|---|
| $\mu$ (cell$^{-1}$.day$^{-1}$) | Dissemination coefficient | 2.12e-11 | 1.48e+03 | 17.3 |
| $\alpha$ (day$^{-1}$) | Gompertzian growth parameter | 1.94 | 18.1 | 2 |
| $\beta$ (day$^{-1}$) | Gompertzian growth parameter | 0.0911 | 19.7 | 2.21 |
| $\lambda$ (day$^{-1}$) | In vitro proliferation rate | 0.837 (fixed) | - | - |
| $k$ (L/mg) | Drug efficacy | 0.446 | 32.1 | 6.34 |
| $k_e$ (day$^{-1}$) | Drug elimination rate | 3.26 (fixed) | - | - |
| $V_d$ (L) | Drug volume of distribution | 12 (fixed) | - | - |
| $\bar{\sigma}$ (%) | Error parameter | 3.91 | - | - |

Abbreviations: CV, coefficient of variation computed as the ratio of the standard deviation and the median of the estimated parameter distribution; r.s.e., residual standard error.

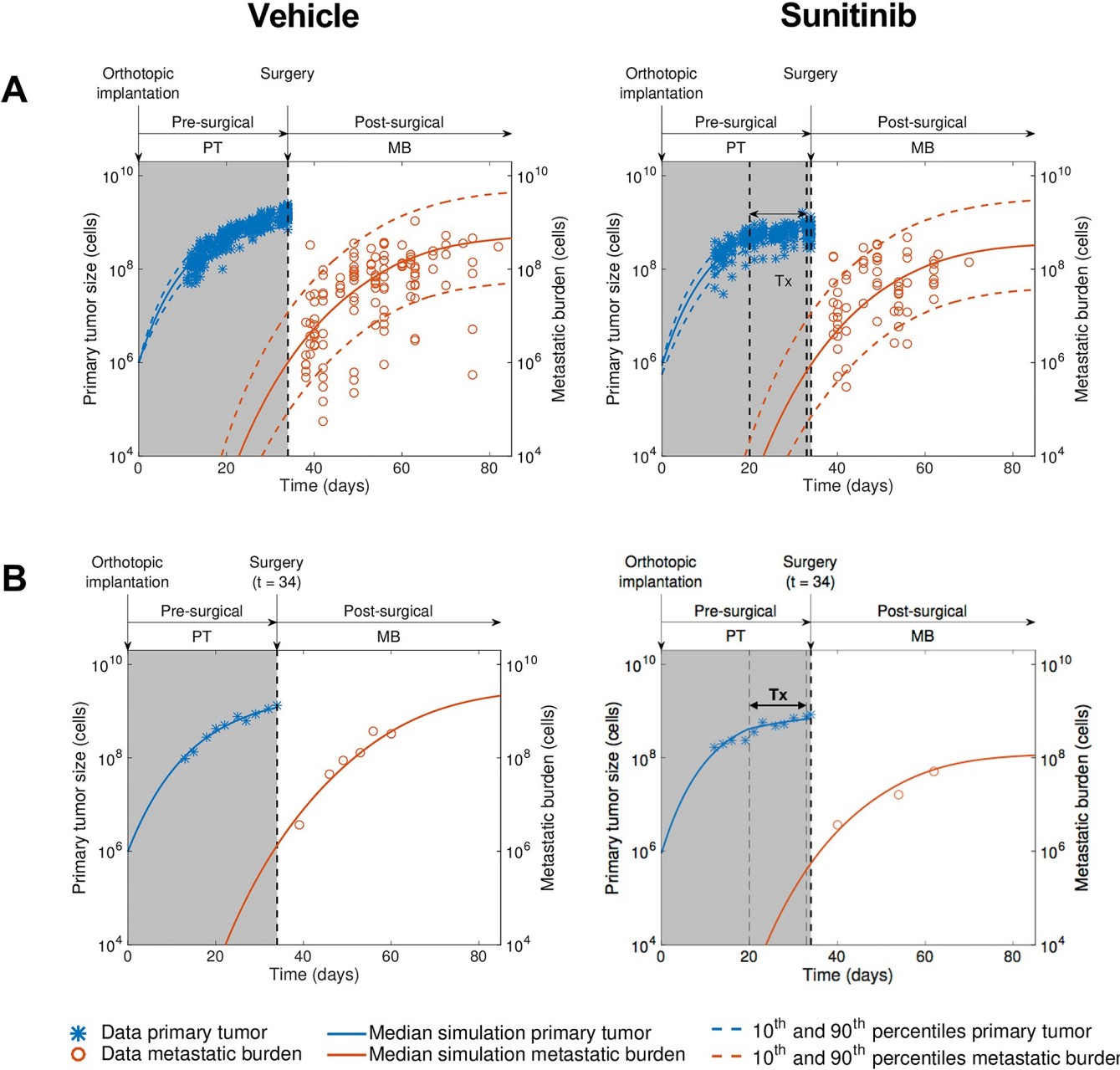

**Fig 2. Calibration and validation of a kinetics-pharmacodynamics (K-PD) mathematical model for neoadjuvant sunitinib treatment effect on pre- and post-surgical tumor growth.** Pre- and postsurgical growth of LM2-4[LUC+] human metastatic breast carcinomas were measured in multiple groups involving different neoadjuvant treatment modalities (doses and durations). The mathematical model was fitted to the experimental data using a mixed-effects population approach (n = 104 animals in total). **(A)** Comparison of the simulated model population distribution (visual predictive check) for vehicle and neoadjuvant sunitinib treatment (60mg/kg/day) 14 days before surgery. **(B)** Examples of individual dynamics. *Tx, treatment; PT, primary tumor; MB, metastatic burden.*

correlations found between the estimated random effects (S7 Fig) confirmed the appropriateness of a full covariance matrix in the assumed distribution of the individual parameters.

Together, these results show the validity of our mathematical model to simulate PT and MB kinetics under a wide range of NAT administration regimens, which can thus be employed to explore *in silico* the quantitative impact of possible NAT schedules.

## Simulations of NAT duration reveal a different impact on PT size reduction and metastasis-free survival

The overall impact of NAT is the combination of i): PT debulking (which in turn reduces metastatic spread from the PT), and ii) an increased risk of metastatic relapse due to delayed removal of the PT. To quantify the impact of NAT duration on these two opposite effects, we ran simulations of our calibrated model for NAT durations ranging from 0 to 18 days. The NAT initiation time was fixed to 27 days after tumor implantation, assumed to be the time of detection of the PT. The resection time was then modulated from 27 to 45 days. Three dose levels were simulated (60, 120, and 240 mg/kg, see Fig 3). First, using only the typical population estimates of the parameters (median individual), we found an important increase in post-surgical MB for long NAT: final values ranged from 2.73 x $10^8$ to 7.52 x $10^8$ cells for NAT durations from 0 to 18 days (176% increase), respectively, at the 60 mg/kg dose level (Fig 3A). This is consistent with our model where NAT does not affect metastatic growth, thus delaying surgery can only increase MB. This was less important in higher dose levels (125% and 48.5% increases for 120 mg/kg and 240 mg/kg, respectively). To study the impact of inter-individual variability, we leveraged our mixed-effects framework to perform population simulations. We simulated 1000 virtual individuals and recorded the percent changes in PT size at the end of NAT, as well as the long-term probability of metastatic relapse. Fig 3B shows the resulting

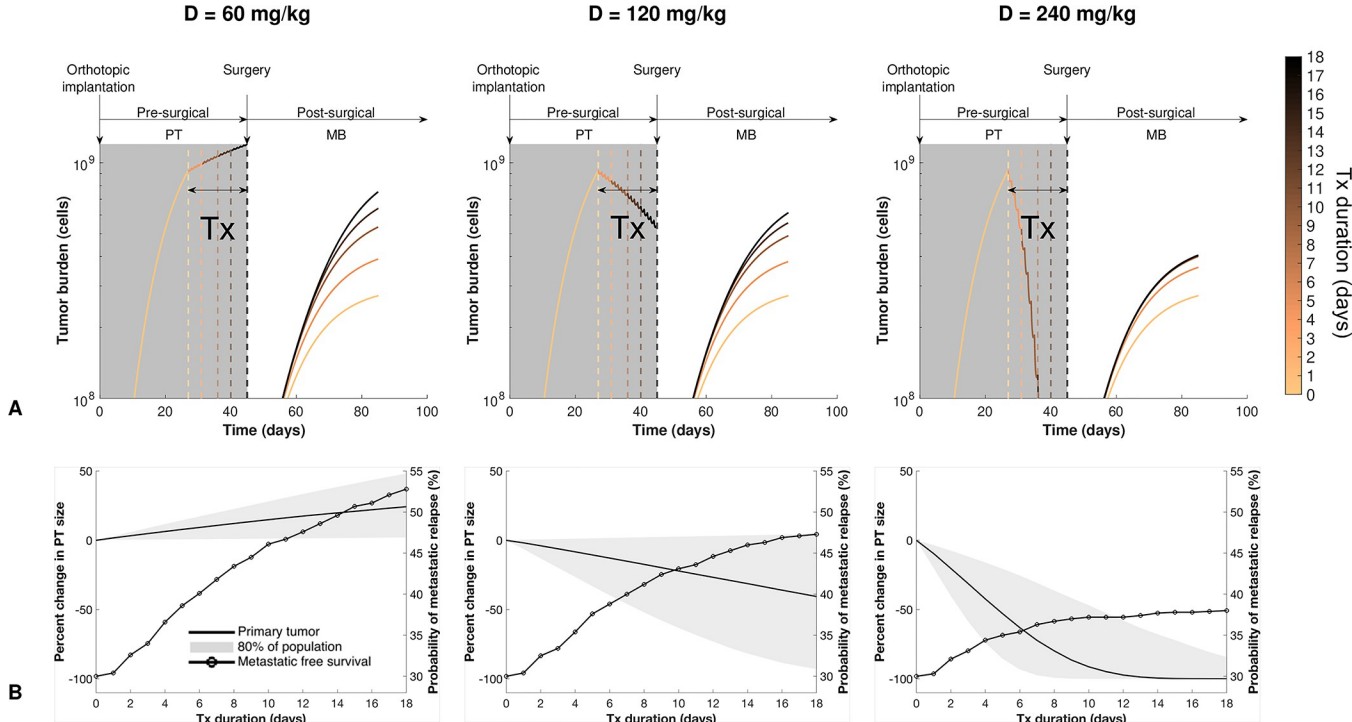

**Fig 3. Simulations of varying neoadjuvant treatment duration quantify contrasted impact on primary tumor size reduction and risk of metastatic relapse.** Using model parameters calibrated from data of our ortho-surgical animal model of breast cancer neoadjuvant targeted treatment, simulations were conducted for treatment durations varying between 0 (light color) and 18 (dark color) days, for three dose levels (60 mg/kg, 120 mg/kg and 240 mg/kg). **(A)** Predicted simulations of pre-surgical primary tumor and post-surgical metastatic kinetics. Note: for the 240 mg/kg plot, the metastatic burden growth curves with the three longest treatment durations are superimposed and not distinguishable. **(B)** Population-level predictions of final primary tumor size (solid line and grey area) and probability of metastatic relapse as functions of the duration of neoadjuvant treatment, which delays surgical removal of the primary tumor (circled line). Inter-individual variability simulated from the population distribution of the parameters learned from the data (n = 1000 virtual subjects). *Tx, treatment; PT, primary tumor; MB, metastatic burden.*

median PT percent changes, together with an area covering 80% of the population, and population metastatic free survival, as functions of NAT duration.

For 60 mg/kg and 120 mg/kg doses, the metastatic relapse risk was predicted to increase drastically when delaying PT removal too long. However, for a 240 mg/kg dose (or for virtual subjects with increased sensitivity to treatment), the increase in metastatic relapse risk was more moderate, since a prolonged NAT was associated with a large decrease in the PT size that translated into a significant reduction of metastatic seeding. Together, these results illustrate how our mathematical model, informed by preclinical data, can provide informative quantitative simulations of the impact of treatment schedules. Our findings suggest a moderate to detrimental impact of long sunitinib NAT at low doses.

## Machine learning for prediction of the metastatic aggressiveness parameter $\mu$ from biomarkers at the time of surgery

Next, we wanted to determine whether biological parameters measured at the time of PT surgery but after NAT could be utilized as predictive biomarkers of postsurgical MB. These biomarkers included immunohistochemical molecular protein measurements of resected PT for cell proliferation (Ki67) and blood vessel (CD31) markers in resected PTs (Fig 4A; example shown), blood-based cellular measurements of circulating myeloid-derived stromal cells (MDSCs) (Fig 4B), and circulating tumor cells (CTCs) from 66 animals (Fig 4C). We investigated whether these molecular and cellular biomarkers may parallel the observed variability in the mathematical parameters, in particular $\mu$, whose large variability indicated potential animal subpopulations of variable metastatic potential values. We first examined correlations among the biomarkers to identify potential redundancies in the data (Fig 4D). High correlations were found between Ki67 and Ki67+/CD31− (r = 0.979, p < $10^{-12}$) and CTC and gMDSC (r = 0.678, p = $3.95 \cdot 10^{-10}$). Next, we investigated the value of these measurements as predictive biomarkers of the mechanistic parameters: $\alpha$ and $\beta$ capture growth kinetics, $k$ the effect of treatment and $\mu$ metastatic dissemination. Fig 4B shows correlations between biomarkers and the parameter estimates. As the individual growth parameters $\alpha$ and $\beta$ were highly correlated (r = 0.997, p < $10^{-5}$), we used the Gompertz tumor doubling time at the volume $V_i = 1$ mm$^3$ to assess the impact of covariates on the tumor growth parameters. It is defined by

$$DT = -\frac{1}{\beta}\ln\left(\frac{\ln(2)+A}{A}\right), \text{ with } A = \ln\left(\frac{V_i}{V_0}\right) - \frac{\alpha}{\beta}.$$ A weak correlation was found between log $(DT)$ and mMDSC levels (Fig 4E, r = 0.275, p = 0.0257). However, none of the available biomarkers was found to correlate either with $\mu$ or log $(\mu)$ (S8 Fig). Next, partial least squares and several machine learning regression algorithms were tested to identify possible relationships between covariates and individual estimates of the metastatic potential parameter (shown in Fig 1A schematic). These included neural networks, support vector machines and random forest models [33]. Cross-validation results for the RMSE of the final regression models were compared against the intercept-only model (the constant model where predictions are the same for all animals, given by the median value in the population, $\mu_{pop}$). As shown in Fig 4F and 4G, none of the fitted models had RMSE or R$^2$ significantly different from the intercept-only model. Values of R$^2$ ranged from 0.133 to 0.199 across the models, with the highest value reached by the conditional random forest model. Prediction error on ln$(\mu)$ ranged from 9.83% ± 10.7% for the best model (conditional random forests, mean ± std) to 10.6% ± 11.3% for the worse (random forests), which was not superior to the predictive power of the intercept-only model (9.71% ± 10.1%). Plotting the observed versus predicted values (S9 Fig) confirmed that the fitted algorithms were unable to explain the variability of parameter $\mu$. Together, these results demonstrate that the biomarkers considered in this study have limited predictive power for metastatic potential as defined by $\mu$.

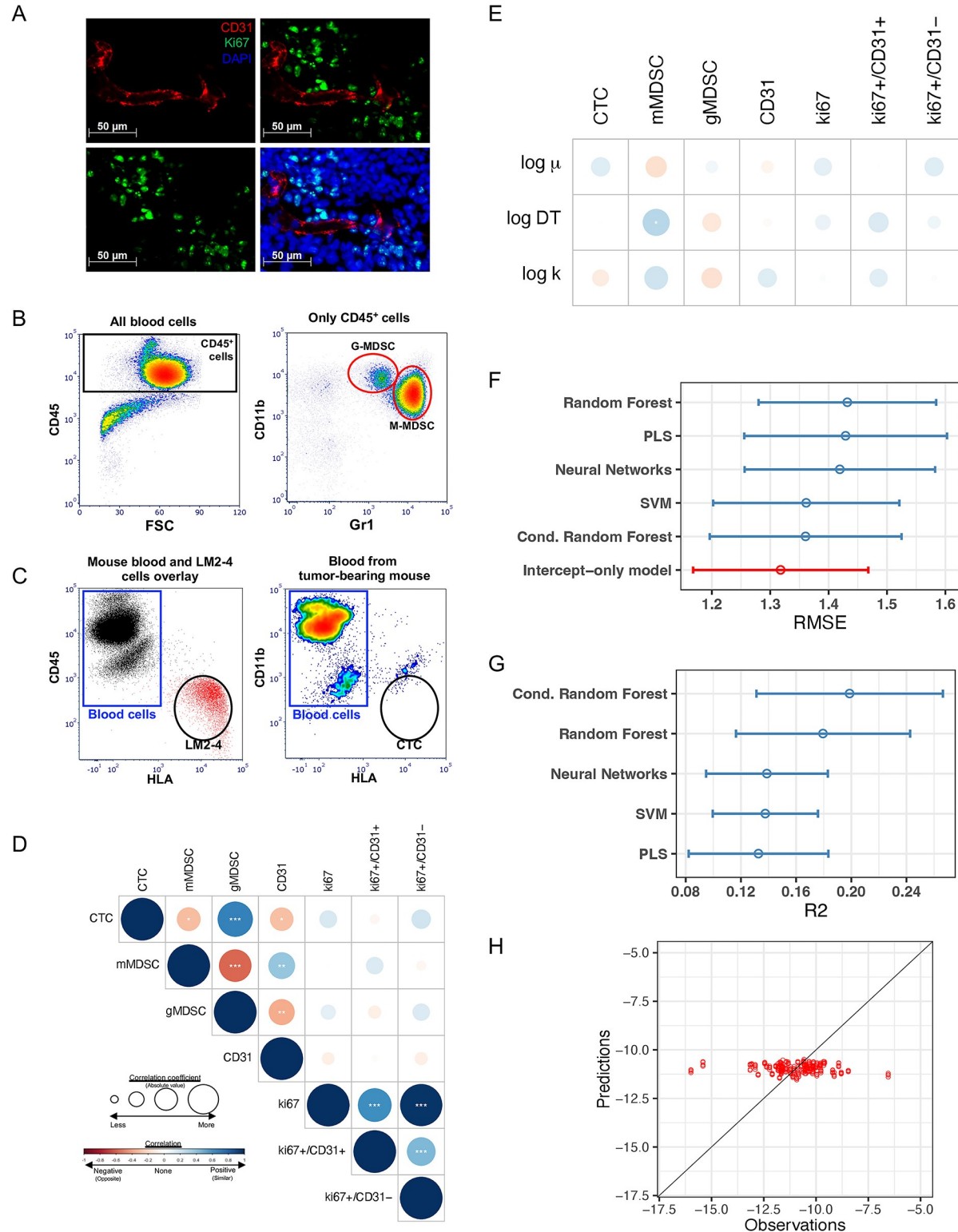

**Fig 4. Use of machine learning algorithms based on presurgical molecular and cellular markers to predict metastatic dissemination parameter 'μ'. (A-C)** Examples of molecular and cellular biomarker analysis. **(A)** Proliferating endothelial cell identification by immunofluorescence. Tissue sections from resected tumors were stained with antibodies against mouse CD31 (red) and mouse Ki67 (green) and counterstained with DAPI (blue). Single channel and merged images are shown. Yellow arrows show proliferating endothelial cells which were counted manually. **(B)** Myeloid-Derived Suppressor Cells (MDSC) quantification by flow cytometry. Whole blood was stained

with anti-mouse antibodies for CD45, CD11b, and Gr1. After selection of CD45-positive cells, MDSCs were analyzed based on CD11b and Gr1 levels. Monocytic-MDSC (M-MDSC) are CD11b+/Gr1 high and granulocytic-MDSC (G-MDSC) are CD11b+/Gr1Medium. Examples of MDSC in untreated and treated animals are shown. **(C)** CTC quantification by flow cytometry. CTCs for xenografts were identified using anti-human HLA. Blood was stained with anti-mouse CD45 and anti-human HLA. Blood and LM2-4 cell samples were overlaid in a dot plot to identify and create the gates for CTCs. Once the gates were created CTC were identified in the blood of tumor-bearing mice. **(D)** Pearson correlation coefficients between biomarkers. Blue (resp. red) color indicates a positive (resp. negative) correlation, with the size of the circle being proportional to the $R^2$ correlation coefficient. * $p<0.05$, ** $p<0.01$, *** $p<0.001$. **(E)** Univariate correlations between the biomarkers and the mathematical parameters. DT = doubling time. **(F)** Cross-validated Root Mean Square Error (RMSE) across different machine learning regression models (see methods) utilizing the values of the biomarkers for predicting $\log(\mu)$. To assess the significance of the covariate in the models, RMSEs were compared against the value of this metric obtained using the intercept-only model. Bars are 95% confidence intervals. Shown in red is the model with the lowest RMSE. PLS = Partial Least Squares. SVM = Support Vector Machines. **(G)** Cross-validated R2 with 95% confidence intervals. **(H)** Predictions versus observations for the conditional random forest algorithm.

## Discussion

A large part of *in vivo* studies in experimental therapeutics focus on the effect of treatments on isolated tumors and few make use of metastatic animal models [34]. However, we and others have previously shown that differential effects occur on the primary tumor and the metastases for some anti-cancer drugs, such as the multitargeted tyrosine kinase inhibitor sunitinib [10,11,35]. Similarly, apart from efforts focusing on evolutionary dynamics of metastasis that do not make use of longitudinal data on size kinetics [36], few quantitative mathematical models exist for metastatic development [4,33,37,38], and none has been quantitatively validated for systemic therapy beyond theoretical considerations [37,39,40]. In previous work we first established such a mathematical model featuring natural metastatic development and surgery of the primary tumor, but no systemic treatment [4]. This was a critical step before being able to model the effect of systemic treatments such as NAT where treatments are limited and long-term benefits are presumed but difficult to quantify because disease recurrence can happen years after surgery, or not recur at all. In the current study, we extended our mathematical model to examine NAT with the RTKI sunitinib leveraging longitudinal data from 128 mice (more than four times more than previous studies [4,38]). Such a large number of subjects and tightly controlled experimental conditions (genetically identical animal background, cell origin, treatment periods, etc.), resulted in precise estimates of the model parameters.

We and others have demonstrated that robust inhibition of primary tumor growth does not always translate into inhibition of metastasis post-surgically nor improvement in survival [10,11,41]. This represents a challenge observed clinically with RTKIs: despite decades of potent tumor-reducing effects in mouse models, efficacy in patients with metastatic disease could be underwhelming. Our mechanistic mathematical modeling approach offers a quantitative tool to simulate and test distinct biological scenarios, in order to distinguish the biological determinants underlying this paradoxical differential effect on the primary and distant tumors. Model predictions (with no fitting involved) allowed to disentangle the impact of NAT on MB of either PT growth arrest alone or PT and metastases growth arrest. They clearly showed: 1) that metastases growth arrest during NAT was very unlikely and 2) that the quantitative impact of PT growth suppression during NAT was sufficient to reproduce the experimental data. Furthermore, NAT exhibited only a negligible reduction in dissemination, as measured by the total MB.

These findings could be explained by the fact that the primary tumor (in the mammary fat pad) and the secondary tumors (mostly in the lungs) would rely on different growth mechanisms, especially at small sizes. Supporting this explanation, a study showed that metastasis relied more on vessel co-option rather than angiogenesis, thus providing them a mechanism of resistance to VEGF RTKI therapy [42]. Beyond NAT, our model predicts limited efficacy of sunitinib in the postsurgical setting, because metastases would likely be similarly small and

rely on similar growth mechanisms. Interestingly, experimental results in mice confirmed this prediction where using a similar metastatic experimental system of triple-negative breast cancer, adjuvant sunitinib did not improve survival [43].

The mechanistic model of NAT validated here provides a valuable tool to explore the impact of the treatment schedules on response and relapse. Simulating varying durations and doses of NAT, we found that long durations of NAT could significantly increase the risk of metastatic relapse when PT response was moderate. Further, our model provides the computational basis to analyze the impact of various NAT dosing regimens in terms of sequence, breaks, and frequency, which is the topic of a companion work.

For breast cancer patients diagnosed with localized disease, predicting the risk and timing of distant metastatic relapse is a major clinical concern [44–46]. Accurate ways to predict the extent of invisible metastatic disease at diagnosis and risk of future metastatic relapse could help to personalize perioperative therapy protocols and avoid highly toxic therapies to patients with low risk of relapse [45]. However, only two risk models [47,48] have met the AJCC criteria for prognostic tools quality so far [49]. Both rely on classical Cox regression survival models. Recently, we have developed a mechanistic approach to metastatic relapse prediction [50]. However, this work did not include the impact of NAT or any systemic treatment. The mathematical model that we validated here on animal data, combined with the methodology developed in [50] lays the groundwork for applications in the clinical NAT setting. It could further refine individual predictions of metastatic relapse in breast cancer by providing surrogate markers of long-term outcomes in addition to pathologic complete response [51]. Indeed, the NAT period represents an invaluable window of opportunity to gather both longitudinal data (such as kinetics of tumor size, pharmacodynamic markers, or circulating DNA from liquid biopsies), as well as single-time biomarkers from the tumor tissue [2]. Here, we propose that mathematical models could form the basis of digital tools able to integrate this multi-parametric and dynamic data into predictive algorithms of both long-term outcome and disease sensitivity to systemic therapy in case of distant relapse.

In the era of artificial intelligence [52], it is to be expected that an increasing number of such prognosis models will appear, combining advances in cancer biology (e.g. molecular gene signatures [45,53]) and imaging [54,55] with algorithmic engineering. Recent years have witnessed the generalization of methods going beyond classical statistical analysis, grouped by the generic term of machine learning (ML) [56]. Here, we proposed an approach to combine ML with mechanistic modeling that consists of using biomarkers at surgery to predict individual mathematical parameters and subsequently postsurgical metastatic evolution. We found overall that the investigated biomarkers contained only limited predictive power of $\mu$, suggesting that alternative biomarkers should be explored in future preclinical and clinical studies. This contrasts with reports showing Ki67 as significantly associated with the risk of metastatic relapse [57]. It might be because Ki67 is a proliferation marker [58], which should rather be predictive of $\alpha$ or the doubling time. Such correlation was observed between Ki67+/CD31+ and DT (Fig 3E), as well as clinical work using our modeling approach [50]. Paired with early clinical trials, our *in vivo/in silico* approach could have translational value to improve biomarkers screening.

Important limitations of our study are that we only analyzed data from one tumor type (triple negative breast cancer), one cell line, one, immune-depressed, animal system, and one drug. This might alter the generalizability and possibly translation of preclinical findings to clinical situations. On the other hand, this is a necessary prerequisite to control as much as possible the heterogeneity in the data, which remained substantial despite a tightly controlled experimental setting. Such conditions ensure robust testing of biological assumptions underlying our mathematical models and, eventually, refutation of unplausible ones (here, that primary and secondary growth would be equally suppressed by NAT).

Given the increasingly diverse arsenal of systemic anti-cancer therapies available with the approval of immune-checkpoint inhibitors, optimal treatment sequence [12,59–61] and dosing regimen [62,63] are becoming crucial issues. Our model could be used and extended to guide the rational design of treatment schedules and modes of combination of immunotherapy with another systemic drug, before preclinical or clinical testing. For immunotherapy, the model would need to be developed further and at least include an additional systemic variable representing the immune system. Immuno-monitoring quantifications could provide an invaluable source of longitudinal data to feed mechanistic models [64]. In addition, response to neoadjuvant therapy could be used to predict which patients are more likely to benefit from adjuvant therapy [9]. Combining artificial intelligence techniques with mechanistic modeling, our modeling methodology offers a way to perform such predictions quantitatively and possibly personalize therapeutic intervention.

## Supporting information

**S1 Table. Neoadjuvant treatment schedules and doses.** Animal groups showing treatment schedules and dosing during a presurgical neoadjuvant period of 14 days.
(PDF)

**S1 Fig. Comparison of simulation of therapy (A) vs no therapy (B) on metastases.**
(PDF)

**S2 Fig. Population fits of all the groups used to calibrate the model parameters (surgery at day 34).**
(PDF)

**S3 Fig. Representative individual fits of the model for Sunitinib-treated animals.**
(PDF)

**S4 Fig. Model predictions in independent datasets (surgery at day 38).**
(PDF)

**S5 Fig. Model diagnostic plots. (A)** Observation vs. individual prediction. Solid lines are identity lines. Dashed lines represent 90% prediction intervals. **(B)** Individual weighted residuals (IWRES) vs time. **(C)** Individual weighted residuals vs log-transformed individual predictions.
(PDF)

**S6 Fig. Distribution of the individual parameters.**
(PDF)

**S7 Fig. Correlations between random effects.**
(PDF)

**S8 Fig. Individual parameters vs covariates.**
(PDF)

**S9 Fig. Observed vs Predicted values for the machine learning algorithms.**
(PDF)

## Author Contributions

**Conceptualization:** Sebastien Benzekry, John M. L. Ebos.

**Data curation:** Michalis Mastri, John M. L. Ebos.

**Formal analysis:** Sebastien Benzekry, Chiara Nicolò.

**Funding acquisition:** Sebastien Benzekry, John M. L. Ebos.

**Investigation:** Sebastien Benzekry, Michalis Mastri, Chiara Nicolò, John M. L. Ebos.

**Methodology:** Sebastien Benzekry, John M. L. Ebos.

**Project administration:** Sebastien Benzekry, John M. L. Ebos.

**Resources:** Sebastien Benzekry.

**Software:** Sebastien Benzekry, Chiara Nicolò.

**Supervision:** Sebastien Benzekry, John M. L. Ebos.

**Validation:** Sebastien Benzekry, John M. L. Ebos.

**Visualization:** Sebastien Benzekry, Michalis Mastri, John M. L. Ebos.

**Writing – original draft:** Sebastien Benzekry, Chiara Nicolò.

**Writing – review & editing:** Sebastien Benzekry, Michalis Mastri, John M. L. Ebos.

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
