## [Decision Letter · Decision Letter 0]

5 Jan 2024

Dear Dr Benzekry,

Thank you very much for submitting your manuscript "Machine-learning and mechanistic modeling of metastatic breast cancer after neoadjuvant treatment" for consideration at PLOS Computational Biology.

As with all papers reviewed by the journal, your manuscript was reviewed by members of the editorial board and by several independent reviewers. In light of the reviews (below this email), we would like to invite the resubmission of a significantly-revised version that takes into account the reviewers' comments.

We cannot make any decision about publication until we have seen the revised manuscript and your response to the reviewers' comments. Your revised manuscript is also likely to be sent to reviewers for further evaluation.

Sincerely,

Philip K Maini

Academic Editor

PLOS Computational Biology

Pedro Mendes

Section Editor

PLOS Computational Biology

Reviewer's Responses to Questions

**Comments to the Authors:**

Reviewer #1: Benzekry et al present a mechanistic modeling of metastatic breast cancer after neoadjuvant treatment with sunitinib that is based on preclinical data using a mouse model of spontaneous metastasis after surgery. This work uses an extension of a previous model developed by this team in untreated animals (Ref 4). The model is sensible and the authors tested whether the neoadjuvant treatment inhibited primary tumor growth as well as post-surgery metastatic growth. In addition they used a machine-learning approach to assess the predictivity of cellular and molecular biomarkers.

The work is interesting and useful. The modeling approach is well done and sound. My main comments aim at improving and clarifying the presentation and discussion of the results.

Major comments:

My major comments refer to the flow and the structure of the paper. Some results are given in the introduction as well as in the methods and some of the methods are in the results. More specifically:

1) The last sentence of the introduction (Lines 50-52) give results and should be removed.

2) Lines 196-198 in the methods give the results that are also presented in the proper section (Lines 263-269 and Fig 1C). Those results should be omitted from the methods.

3) The first section of the result should be deleted and content moved to the introduction and/or the methods as appropriate. Lines 235-242 already are in the methods (Lines 152…) or in the introduction (Lines 18-24). Lines 242-246 and Fig 1A describe the experimental protocol and should be in the methods.

4) In the results, Lines 250-254 describe previous results and should be either in the introduction or even better in the discussion. It is actually unclear what the current results bring with respect to those previous results in term of the “differential effects with suppression of presurgical PT growth not consistently translating into reduction in postsurgical MB” as described later in this section (Lines 266-269). This should be discussed.

5) Table 1 reports parameter estimates for a survival model. Is this the model used to predict metastatic free survival in Fig 3B? The survival model is not described in the method section. It should be.

Other comments:

6) Lines 4-5: It is not the neoadjuvant trials per se that “provide several advantages to assist in novel drug development and translational research” but rather model-based analyses of those trials. The trials are designed to address unmet medical needs.

7) Lines 35-50: These considerations should go in the discussion. The introduction should stop at line 34.

8) Line 352: Is 4B the right figure? Or is it 4E?

9) Line 372: Fig 5 is missing. It is probably not needed, Fig S9 is fine.

10) The discussion is a bit long and focuses too much on the machine learning part of the investigation… which in the end did not generate any predictive marker. The reviewer suggest to shorten this part.

11) Lines 399-403: This sentence suggests that Ref 13 is from the same team as the authors when it does not seem to be the case?

12 Lines 459-461: Are there any limitations to such investigations in preclinical models to predict clinical situations? This point may be discussed.

13) Lines 475-483 seem to be describing another study that is not presented in this paper. It does not bring anything to the current discussion. The discussion being quite lengthy (see comment 10), I suggest to remove.

Reviewer #2: Review of “Machine-learning and mechanistic modeling of metastatic breast cancer after neoadjuvant treatment”,

submitted by Benzekry et al. to PLOS Computational Biology

The manuscript under review presents a combined experimental and computational study of the effect of neoadjuvant therapy using the tyrosine kinase inhibitor sunitinib in a surgical metastatic breast cancer mouse model. The study is very comprehensive, with 120 animals in total and a variety of different treatment/surgery schedules. First, parameters of a previously published computational model for primary and metastatic tumor growth are estimated in a mixed-effects framework. It is found that sunitinib treatment only affects the primary tumor, but not metastases. Subsequently, the authors employ various machine learning models to investigate the relationship between biomarkers determined at surgical removal of the primary tumor and individual estimates of a parameter quantifying metastatic spreading. They cannot establish any significant relationships and conclude that this preclinical model can serve as a pre-screening step for investigations of (here, unsuccessful) biomarkers in neoadjuvant trials. The study is innovative, comprehensive and has clear translational orientation towards a relevant clinical application. However, there are still some open questions and technical details to be addressed, as outlined below. Therefore, I think that the manuscript is in principle well suited for publication in PLOS Computational Biology, but some revisions are necessary first.

Comment 1

The authors mention a survival model in the Caption of Fig 1 and in Table 1. However, it is not discussed or mentioned anywhere else. It should be part of the Methods section (including the type of hazard distribution used) and results should be discussed.

Comment 2

Before arriving at their metastatic model in which therapy affects the primary tumor but not metastases (model B), they investigated a model in which primary and secondary tumors are equally affected (model A). It would be instructive to investigate an intermediate model that has some effect on metastases, but not as much as the primary tumor (i.e., drug efficacy would be reduced by some factor to be estimated). In fact, although model B clearly fits the data much better than model A in Fig S1, in some panels it slightly overpredicts metastatic burden at the early timepoints (~40 days). Beyond this observation, it would be conceptually important to consider (and possibly reject) such an intermediate model, since the finding that neoadjuvant treatment always has negative consequences on metastatic relapse relies on the structure of model B.

Comment 3

Regarding the use of machine learning to link biomarkers to predict individual model parameters, the authors currently only have a negative example with parameter mu. However, in L461-464 they also discuss a scenario with a potentially positive finding, namely the link between Ki67 and metastatic relapse. They stated that this link could be mediated through another parameter, namely alpha, which should also impact on the metastatic relapse metric that is being used. Why did the authors only try to predict mu and not alpha? Presenting a positive result would substantiate their claim of translational relevance further.

Comment 4

The authors briefly introduce their concept of metastatic relapse in the Results section (L319-320). After reading this part, I had no intuition about which metastatic burdens would be considered a relapse or not. Hence, the definition should be moved to the Methods section and contain additional details, in particular the cutoff value in terms of metastatic burden that results from their rationale. Potentially, this cutoff could even be integrated in some of the figures.

Comment 5

In the caption of Fig. 1C, it is stated that simulations were done with a previously estimated set of model parameters based on a dataset of untreated animals. How can a treatment be simulated using such parameters? At least the drug efficacy parameter k must have been taken from somewhere else.

Comment 6

I don’t understand Fig 3A. “Tx” seems to be constant, while “NAT duration” changes from 0 to 18 days. First, only one term should be used for treatment (say it’s NAT), consistently throughout the manuscript. Next, a varying treatment duration should result either in a varying start of treatment or varying time of surgery (it is not currently stated which of the two it is).

Errors, typos and suggestions for phrasing

• The second to last sentence in the abstract is not that clear, it is better phrased at the end of the introduction (L50-52).

• In the Major findings section, the term RTKI is not explained. In fact, the abbreviation is not explained anywhere, it rather seems to be used synonymously for “VEGFR TKI”. Please stick to only one of these terms.

• In the first paragraph of the Introduction, two topics are mixed up, namely neoadjuvant trials and breast cancer. Please introduce them separately first, and only then discuss neoadjuvant trials for breast cancer.

• L 5: “faster to conduct” compared to what?

• L24 remove “which we termed mu”, it is not helpful in the introduction.

• L26 it should be “hypotheses”, not “hypothesis”

• L89 “tumor” missing after “primary”

• L149 it should be Ki67 instead of Ki76

• L157, in the second argument of the min function, “*v” is missing.

• L208 use “model prediction” instead of “model value”

• The error models in L 210 and L 215 contradict each other. Please rewrite or remove L 210 for consistency.

• L405 do you mean “could be considered underwhelming”?

• L408: I was irritated by the phrasing “with no fitting involved” since in L201-203, it is stated that model parameters were estimated using a mixed-effect model.

• There’s an error in the middle panel of Fig 3. Surgery should be at the second, not first dashed line.

• Also in Fig 3, color coding of axes and lines would make panel B easier to read.

• Residual variability parameters for primary tumor and metastases are missing in Table 1

• Caption Tab S1, “Doeses” should be “Doses”

• In Fig S5, please explain the color-coding: left column primary tumor, right column metastases (even though it is used consistently everywhere).

**Have the authors made all data and (if applicable) computational code underlying the findings in their manuscript fully available?**

Reviewer #1: **No: **The authors did nor share the code for the model

Reviewer #2: **No: **In their Data and Code Availability statement, the authors claim that all relevant data are provided in the manuscript. However, neither the MATLAB code used for the analysis nor any of the experimental data are provided, and there is no reference in the text or supplement as to where they could be found.

PLOS authors have the option to publish the peer review history of their article (what does this mean?). If published, this will include your full peer review and any attached files.

Reviewer #1: No

Reviewer #2: No
---

## [Decision Letter · Decision Letter 1]

22 Feb 2024

Dear Dr Benzekry,

Thank you very much for submitting your manuscript "Machine-learning and mechanistic modeling of metastatic breast cancer after neoadjuvant treatment" for consideration at PLOS Computational Biology. As with all papers reviewed by the journal, your manuscript was reviewed by members of the editorial board and by several independent reviewers. The reviewers appreciated the attention to an important topic. Based on the reviews, we are likely to accept this manuscript for publication, providing that you modify the manuscript according to the review recommendations.

Sincerely,

Philip K Maini

Academic Editor

PLOS Computational Biology

Pedro Mendes

Section Editor

PLOS Computational Biology

Reviewer's Responses to Questions

**Comments to the Authors:**

Reviewer #1: The authors addressed my concerns. No further comment.

Reviewer #2: The authors submitted a revised version that answered most of my requests. However, there are still some two issues that need to be addressed.

Comment 4

The authors have moved their definition of metastatic relapse to the Methods section as requested, but unfortunately, they didn’t address the second part of my request. It is still not stated which threshold value (in terms of metastatic burden) results from their rationale. All I can see is that, from Fig S2 (Vehicle), the threshold is somewhere around 10^8 cells. But since only 10th, 50th and 90th percentiles are drawn in Fig S2 (and no 30th percentile, which enters the relapse definition), this can only be guessed. Please add the exact numeric threshold value to the Results.

Comment 6

Unfortunately, the explanations and revisions regarding this point are not at all satisfying. The authors merely changed the word “Tx” to “NATT” in Figure 3 without addressing my concerns. First, if, as the authors state in the rebuttal, the time of surgery is fixed at 45 days, then the gradient in the metastatic burden as a function of NATT should be inverted, i.e. higher burden for NATT = 0 days than for 18 days. Second, it is very confusing to present a figure in which the primary tumor burden is shown for one NATT duration, while the metastatic burden is shown for different ones, in the same panel. There should be one PT burden curve per NATT duration, decreasing from different time points onwards. Taking these two points together, it seems possible that the simulations were carried out in a different way: fixed start of NATT, but different times of surgery. This would explain both the gradient in metastatic burden and the overlapping primary tumor burden. However, even in this scenario, there should be some graphical hint in the left part of the panels that NATT durations differ. Finally, annotations are still not used consistently. In the legend of Fig 3B, a reference is made to the survival model. Also, the terms Tx / NAT / NATT are still mixed up throughout the manuscript/captions/figures.

**Have the authors made all data and (if applicable) computational code underlying the findings in their manuscript fully available?**

Reviewer #1: Yes

Reviewer #2: **No: **The authors provided a Zenodo DOI for the underlying experimental data, but not for the computational code used in the analysis. Therefore, I have not been able to verify if the computations were carried out as described in the manuscript.

PLOS authors have the option to publish the peer review history of their article (what does this mean?). If published, this will include your full peer review and any attached files.

Reviewer #1: No

Reviewer #2: No

Figure Files:

Data Requirements:

Reproducibility:

References:

---

## [Decision Letter · Decision Letter 2]

26 Mar 2024

Dear Dr Benzekry,

Thank you very much for submitting your manuscript "Machine-learning and mechanistic modeling of metastatic breast cancer after neoadjuvant treatment" for consideration at PLOS Computational Biology. As with all papers reviewed by the journal, your manuscript was reviewed by members of the editorial board and by several independent reviewers. The reviewers appreciated the attention to an important topic. Based on the reviews, we are likely to accept this manuscript for publication, providing that you modify the manuscript according to the review recommendations.

Please see issue regarding availability of the code.

Sincerely,

Philip K Maini

Academic Editor

PLOS Computational Biology

Pedro Mendes

Section Editor

PLOS Computational Biology

Reviewer's Responses to Questions

**Comments to the Authors:**

Reviewer #2: The authors have appropriately addressed the two remaining issues that I raised in the last round of revision. However, I still cannot see any computational code submission -- only the experimental data are stored under the Zenodo link.

**Have the authors made all data and (if applicable) computational code underlying the findings in their manuscript fully available?**

Reviewer #2: **No: **The authors provide a Zenodo link where all experimental data are stored. However, the computational code they used for data analysis is still not made available.

PLOS authors have the option to publish the peer review history of their article (what does this mean?). If published, this will include your full peer review and any attached files.

Reviewer #2: No

Figure Files:

Data Requirements:

Reproducibility:

References:

---

## [Editor Report · Decision Letter 3]

18 Apr 2024

Dear Dr Benzekry,

We are pleased to inform you that your manuscript 'Machine-learning and mechanistic modeling of metastatic breast cancer after neoadjuvant treatment' has been provisionally accepted for publication in PLOS Computational Biology.

Best regards,

Philip K Maini

Academic Editor

PLOS Computational Biology

Pedro Mendes

Section Editor

PLOS Computational Biology

---

## [Editor Report · Acceptance letter]

23 Apr 2024

PCOMPBIOL-D-23-01812R3 

Machine-learning and mechanistic modeling of metastatic breast cancer after neoadjuvant treatment

Dear Dr Benzekry,

I am pleased to inform you that your manuscript has been formally accepted for publication in PLOS Computational Biology. Your manuscript is now with our production department and you will be notified of the publication date in due course.

With kind regards,

Zsofia Freund
